# Colchicine for Prevention of Atrial Fibrillation after Cardiac Surgery in the Early Postoperative Period

**DOI:** 10.3390/jcm11051387

**Published:** 2022-03-03

**Authors:** Vladimir Shvartz, Tatyana Le, Yuri Kryukov, Maria Sokolskaya, Artak Ispiryan, Eleonora Khugaeva, Gulsuna Yurkulieva, Elena Shvartz, Andrey Petrosyan, Leo Bockeria, Olga Bockeria

**Affiliations:** 1Department of Surgical Treatment for Interactive Pathology, Bakoulev Scientific Center for Cardiovascular Surgery, 121552 Moscow, Russia; tgle@bakulev.ru (T.L.); masokolskaya@bakulev.ru (M.S.); ayispiryan@bakulev.ru (A.I.); doc.khugaeva@mail.ru (E.K.); yurkulieva1991gulya@mail.ru (G.Y.); adpetrosyan@bakulev.ru (A.P.); bockeria@bakulev.ru (L.B.); soleo2003@gmail.com (O.B.); 2Department of Cardiovascular Surgery, Arrhythmology and Clinical Electrophysiology, Bakoulev Scientific Center for Cardiovascular Surgery, 121552 Moscow, Russia; yurijkuban@mail.ru; 3National Medical Research Center for Therapy and Preventive Medicine, 101990 Moscow, Russia; shvartz.en@ya.ru

**Keywords:** colchicine, postoperative atrial fibrillation, coronary artery bypass grafting, aortic valve replacement

## Abstract

Background. Postoperative atrial fibrillation (POAF) is a common complication of cardiac surgery. It has been proven to be associated with an increase in the incidence of early complications and mortality, an increase in the rate of hospital stay duration, and economic costs of their treatment. One of the pharmaceutical drugs recommended by the American College of Cardiology (ACC)/American Heart Association (AHA) for preventing POAF is colchicine (class IIB). However, the results of research on the efficacy and safety of colchicine are ambiguous and, consequently, require further study. Objective. Evaluating the efficacy of short-term colchicine administration in the prevention of POAF in patients after open-heart surgery. Materials and methods. Double-blind, randomized, placebo-controlled clinical trial. The subjects were randomly assigned to two groups: treatment group (n = 50) with subjects receiving 1 mg of colchicine 24 h before the surgery, as well as on days 2, 3, 4, and 5 in the postoperative period; and the control group (n = 51), receiving placebo on the same schedule. The primary endpoint was the frequency of POAF in both groups within 7 days after surgery. Results. The study included 101 patients (82 men, 19 women). Baseline clinical, laboratory, instrumental, and intraoperative data did not differ statistically significantly between the groups. POAF was detected in 9 patients (18%) of the treatment group and 15 subjects (29.4%) of the control group, which had no statistical significance (odds ratio, OR 0.527; 95% Cl 0.206–1.349; *p* = 0.178). No statistically significant differences were revealed for most secondary endpoints, as well as between the groups for all laboratory parameters. There were statistically significant differences between the groups solely in the frequency of diarrhea: 16 (32%) patients in the treatment group and 6 (11.8%) subjects in the control group (OR 3.529; 95% Cl 1.249–9.972; *p* = 0.010). Conclusions. We did not detect any statistical differences between the groups in terms of primary endpoints, which could be due to the insufficient volume of the sample of the study. However, we detected some trends of statistical differences among the groups in terms of some parameters. Clinical Trials Registration. ClinicalTrials. Unique identifier: NCT04224545.

## 1. Introduction

Postoperative atrial fibrillation (POAF) is a frequent (20–40%) complication of cardiac surgery, which has been proven to be associated with an increase in the incidence of early complications and mortality, as well as an increase in the hospital stay duration of the patients and economic costs of their treatment [1,2,3,4,5]. The annual increase in the number of heart surgeries makes this problem extremely urgent. The prevention of POAF is an important component in treating this category of patients. To achieve this goal, antiarrhythmic drugs (amiodarone, beta-blockers) are conventionally used [6,7]. In some studies, anti-inflammatory drugs (NSAIDs, corticosteroids, statins) have been seen to have a positive effect on the frequency of POAF, thereby proving a certain role of inflammatory processes in the pathogenesis of POAF [8,9]. Colchicine, with its anti-inflammatory properties, may be effective in POAF prevention [10,11,12,13,14]. According to American College of Cardiology (ACC)/American Heart Association (AHA) clinical guidelines, the use of colchicine has an evidence class IIB [15,16], “*usefulness/efficacy is less well established by evidence/opinion*”.

Two consecutive studies, COPPS-1 and COPPS-2, assessed the effect of colchicine on the incidence of POAF after open heart surgery (COPPS-1) and the development of postcardiotomy syndrome (COPPS-2) [12,13]. In COPPS-1, subjects received colchicine from day 3 after surgery for a month and exhibited a significant decrease in POAF after open-heart surgery. However, in COPPS-2, colchicine did not significantly affect the incidence of POAF. Based on the results of a meta-analysis, a decrease in POAF incidence in the treatment group versus control group was established (hazard ratio, HR = 0.69, 95% CI 0.57–0.84, *p* = 0.0002) [17].

Therefore, the outcomes of research on the efficacy of colchicine are ambiguous and require further investigation [18,19,20]. In this regard, the objective of our study was to evaluate the efficacy of short-term colchicine administration in terms of preventing POAF in patients after open-heart surgery.

## 2. Materials and Method

### 2.1. Study Design

Double-blind, randomized, placebo-controlled clinical trial, Colchicine in Cardiac Surgery (COCS). http://clinicaltrials.gov (accessed on 3 February 2022): unique identifier—NCT04224545. Our study protocol complied with the ethical guidelines of the 1975 Declaration of Helsinki and with the Ethical Guidelines for Epidemiological Research by the Russian Federation Government. The study was approved by the Local Ethics Committee of Bakulev Center for Cardiovascular Surgery of the Ministry of Healthcare of the Russian Federation on 20 December 2019 (Protocol #2 date 20 December 2019). Written informed consent was obtained from each prospective participant prior to the randomization procedure.

The study was carried out at the Department of Surgical Treatment of Interactive Pathology of Bakulev Center for Cardiovascular Surgery of the Ministry of Healthcare of the Russian Federation.

### 2.2. Inclusion Criteria

The study included subjects 40–80 years old who were scheduled for coronary artery bypass grafting (CABG) and/or aortic valve replacement (AVR).

### 2.3. Exclusion Criteria

The exclusion criteria were as follows: any form of atrial fibrillation or atrial flutter in anamnesis; congenital heart defects except for bicuspid aortic valve; frequent ventricular or supraventricular extrasystoles, 2nd and 3rd degree 2–3 AV block; intake of glucocorticosteroids or any antiarrhythmics, except beta blockers, within a month prior to the surgery; previous open heart; moderate or severe chronic renal failure (creatinine clearance less than 50 mL/min); liver disease; mitral valve pathology (insufficiency and/or stenosis grade 2 or higher); patient’s involvement in another clinical research. The reasons for withdrawal from the study after randomization were: hospital death on day 1 after surgery, extended stay (longer than 1 day) at the resuscitation department or intensive care unit after the surgery, and a patient’s desire to stop participating in the study.

### 2.4. Randomization

Patients were randomly assigned to two groups: treatment group (n = 50, receiving 1 mg of colchicine 24 h. before the surgery, as well as on days 2, 3, 4, and 5 in the postoperative period in conjunction with optimal medicamentous therapy) and the control group (n = 51, receiving placebo on the same schedule). We employed limited block randomization, with a block size of 20.

### 2.5. Statistical Analysis

Statistical analysis was performed using the SPSS^®^ Statistics 20.0 software (IBM, Armonk, NY, USA). Data are presented as median and interquartile range: Me (Q1; Q3). To compare two independent samples, we used the Mann–Whitney U test for quantitative variables; Pearson’s chi-square test or Fisher’s exact test were used for categorical variables. The difference between the groups was considered statistically significant at *p* < 0.05.

### 2.6. Endpoints: Measuring Primary Outcome

The primary endpoint was the frequency of POAF in groups within 7 days after surgery. The episode with the absence of visible regular p-waves, the appearance of f-waves and irregular R-R intervals on the ECG for more than 5 min was the diagnostic confirmation of POAF.

### 2.7. Endpoints: Measuring Secondary Outcomes

The secondary endpoints were the frequency of lethal and nonlethal events, pericardial effusion, pleural effusion, acute kidney injury (AKI), plasma inflammation, and liver damage within 7 days after surgery. The main lethal and non-lethal nosocomial events were death, stroke, myocardial infarction, and congestive heart failure. The pericardial and pleural effusion was assessed by the echocardiography. The definition of AKI was based on KDIGO criteria (occurrence of AKI was defined as either an increase in serum creatinine ≥0.3 mg/dL within 48 h or an increase to ≥1.5 times baseline within 7 days). The marker of plasma inflammation was the number of leukocytes and neutrophils. The liver damage was assessed by the level of aspartate aminotransferase (AST) and alanine aminotransferase (ALT).

### 2.8. Surgery

Coronary artery bypass grafting was performed on a beating heart, under the conditions of cardiopulmonary bypass (parallel perfusion) or without cardiopulmonary bypass (off-pump CABG), depending on the preferences of the operating surgeon. The left internal thoracic artery (LITA) with bypass of the anterior interventricular artery (AIA) and great saphenous vein (GSV) were conventionally used as conduits for bypassing the remaining basins of the coronary arteries. In cases of conduit deficiency (history of phlebectomy, GSV varicose), the radial artery was used. In cases when CABG was combined with AVR, the first stage was the collection of conduits in the planned amount, after which the aortic valve was replaced. Then, after the restoration of aortic integrity and integrity of the right atrium, the patient’s body was warmed up to 36.6 °C, and the cardiac activity was reestablished. Next, myocardial revascularization was performed on the beating heart under the condition of cardiopulmonary bypass. Bypass grafting of the target coronary arteries was performed via imposing the distal anastomosis, followed by forming proximal anastomoses with the parietally deflated aorta. Quality control of the formed anastomoses was carried out using intraoperative grafting, which ensured timely detection and elimination of their defects.

### 2.9. Monitoring

Patients were subjected to continuous ECG monitoring: 3-channel ECG monitoring at intensive care units on postoperative days 1 and 2; 10-min 12-channel ECG recording on postoperative days 3 and 7; 24 h ECG monitoring sensu Holter was performed on days 3 and 5 after the surgery. Moreover, on postoperative days 3 and 5, all participants underwent echocardiography and laboratory tests (complete blood count, biochemical blood test). The fact of POAF development was defined as an episode of atrial fibrillation lasting over 5 min. On day 7 after surgery, adverse early postoperative clinical events (nausea, diarrhea, etc.) were noted by interviewing the patients.

## 3. Results

A total of 107 patients were enrolled by means of the randomized sampling procedure, of which six dropped out during our study: three patients from the treatment group and three from the control group (Figure 1). In the treatment group, all patients stopped participating because of their extended stay at the intensive care unit. The one patient in the control group dropped out from the study due to extended stay at the intensive care unit, whereas one more patient died on the first day, and another patient wished to cancel his participation in the study.

As a result, 101 patients were included in the analysis (82 men, 19 women). Background clinical, laboratory (instrumental), and intraoperative data, along with medicamentous therapy did not differ significantly between the groups (Table 1 and Table 2).

In the treatment group, POAF was detected in 9 (18%) patients, while in the control group, it was found in 15 (29.4%) patients. However, this difference was not statistically significant (OR 0.527; 95% Cl 0.206–1.349; *p* = 0.178) (Table 3).

Moreover, we discovered no statistically significant differences among the following secondary endpoints: (1) The presence of pericardial effusion in the treatment group versus the control group: 3 (6%) vs. 8 (15.7%) patients on day 3 after the surgery (OR 0.343; 95% Cl 0.085–1.377; *p* = 0.204), and 8 (16%) vs. 15 (29.4%) subjects on day 5 (OR 0.457; 95% Cl 0.174–1.202; *p* = 0.094), correspondingly; (2) Pleural effusion in the treatment group versus the control group: 22 (44%) vs. 6 (31.4%) subjects on day 3 after the surgery (OR 1.719; 95% Cl 0.762–3.887; *p* = 0.139), and 19 (38%) vs. 21 (41.2%) patients on day 5 (OR 0.876; 95% Cl 0.394–1.945; *p* = 0.809), respectively. Acute kidney injury (AKI) on day 3 was noted in 1 patient in the treatment group and 3 patients in the control group (OR 0.327; 95% Cl 0.033–3.250; *p* = 0.618). On day 5, there was 1 AKI subject in the treatment group and none in the control group (*p* = 1.000). Both differences were not statistically significant. Besides, there were no lethal outcomes. (Table 3 and Table 4).

There were no statistically significant differences between the groups on days 3 and 5 in terms of all laboratory parameters (including the content of neutrophils, ALT, and AST) (Table 5). Medicamentous therapy in the postoperative period did not differ as well (Table 6).

Among the adverse events, statistically significant differences between the groups were revealed only for the occurrence of diarrhea in the early postoperative period: 16 (32%) patients in the treatment group vs. 6 (11.8%) patients in the control group (OR 3.529; 95% Cl 1.249–9.972; *p* = 0.010) (Table 7).

## 4. Discussion

We conducted a preliminary analysis of the collected data, which yielded no statistical differences between the groups on the primary endpoint and most secondary endpoints.

A statistically significant difference was discovered only for the frequency of diarrhea occurrence in the early postoperative period. This indicator was higher in the treatment group (16 (32%) patients) than in the control group (6 (11.8%) patients), which was consistent with available published data on the negative effect of colchicine on the digestive tract [6,7,10,19].

It should also be noted that trends of approaching statistical differences were discovered for some parameters. This finding probably implies a lack of representativeness for the sample under study and a lack of statistical power.

In COPPS-1, patients showed a significant decrease in POPF after open-heart surgery. However, in COPPS-2, colchicine did not have a significant effect on the incidence of POAF [12,13]. Based on the results of the meta-analysis, a decrease in the frequency of POAF was revealed in the group of patients receiving colchicine compared with the control group.

Contradictory research results may be associated with different inclusion and exclusion criteria, the duration of colchicine intake, the period of patient follow-up, and the diagnostic criteria of POAF.

In the COPPS-2 study [13], colchicine was administered within 48–72 h before surgery and within 1 month after surgery. In our study, patients take colchicine once a day before surgery and on days 2 and 5 after surgery. The primary endpoint in the COPPS-2 study was an episode of POAF that lasted at least 30 s, recorded within 3 months after surgery. In our study, the observation of patients was limited to seven postoperative days, and the diagnostic criteria for AF was an episode of AF lasted at least five min. This really affects the different frequency of primary endpoint detection in our previous studies. It is logical that the stricter the criteria for the diagnosis of POAF, the frequency of detection of the event will be higher. However, at the moment, there are no clear universal criteria for determining clinical POAF (30 s or 5 min). There are several studies where POAF has been defined as AF lasting 10 min or more. In addition, there is no reliable data on prognostic differences in events with different interpretations of the definition of POAF.

In the COPPS-1 study [12], colchicine intake was carried out on the 3rd day after surgery and lasted for 1 month after surgery. The patients with episodes of AF registered on the first and second postoperative days were excluded from the analysis, while the episode of AF with duration at least of five min as well as in our study was considered to be the diagnostic criteria.

Furthermore, in the COPPS-1 and COPPS-2 studies, patients with a history of AF, taking antiarrhythmic drugs in the preoperative period, mitral valve defects, which have been proven to increase the risk of AF, were not excluded initially from the study. In addition, these studies included patients who underwent CABG, operations on various heart valves and aorta. Our study included patients only after CABG surgery and/or replacement of aortic valve (without significant mitral pathology), and the atrial fibrillation or atrial flutter in history was the exclusion criteria for our study. Thus, we initially excluded patients with a high risk of developing of AF in order to minimize the influence of existing predictors of occurrence of AF in the preoperative period. We consider that our cohort was more homogeneous initially, although it was smaller than in previous studies.

In fact, the protocol of our study differs significantly from previous studies by the more homogeneous group of patients and the absence of high-risk factors for the development of AF (previous AF in the history, mitral valve interventions, etc.). The obtained results and data should be used for subsequent meta-analyses and for calculating the required number of patients when planning larger randomized trials in the future. The datasets analyzed during the current study are publicly available. The data is available in the general repository “Open Science Framework” at the link https://doi.org/10.17605/OSF.IO/HPCZW (accessed on 3 February 2022).

### Limitations of the Study

In this study, there was a small sample of patients, although it was more homogeneous than in previous studies.

In addition, some patients took NSAIDs in the postoperative period, which could have affected the results.

For the assessment of inflammation in the postoperative period, the levels of leukocytes and neutrophils in the blood were studied as markers of inflammation, although the markers of the interleukin family (IL-1, IL-6, IL-10), TNFa, NLRP3, etc. are more specific.

## 5. Conclusions

Data have been obtained on the efficacy of short-term colchicine administration aimed at preventing POAF after undergoing CABG and/or AVR. The frequency of POAF in the treatment group was 18% versus 29.4% in the control group, but the detected differences were not statistically significant (OR 0.527; 95% Cl 0.206–1.349; *p* = 0.178), which could be due to the insufficient volume of the sample of the study.

## Figures and Tables

**Figure 1 jcm-11-01387-f001:**
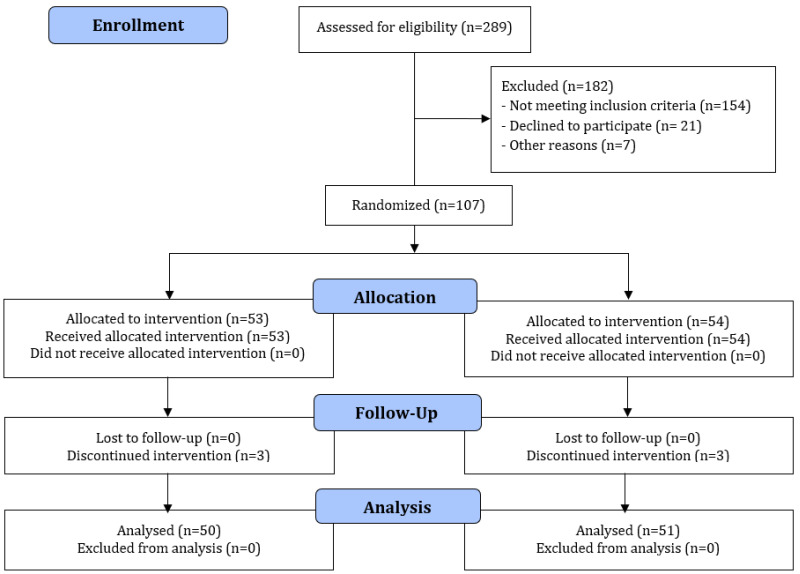
CONSORT diagram.

**Table 1 jcm-11-01387-t001:** Basic parameters of patients, according to the initial data.

Parameters	Colchicine (n = 50)	Placebo (n = 51)	*p*
Clinical parameters of patients			
Age, y	63 (56; 68)	60 (55; 67)	0.420
Male, n (%)	41 (82)	41 (80,4)	1.000
BSA, m^2^	2.01 (1.9; 2.18)	2.04 (1.87; 2.14)	0.922
Weight, kg	85 (74; 94.3)	87 (75; 95)	0.984
BMI, kg/m^2^	29.3 (26.6; 32.1)	28.4 (24.7; 32.4)	0.561
Angina pectoris, n (%)	47 (94)	47 (92.2)	1.000
Angina pectoris class III-IV, n (%)	30 (60)	28 (54.9)	0.604
Diabetes, n (%)	14 (28)	9 (17.6)	0.215
COPD, n (%)	2 (4)	2 (3.9)	1.000
Hypertension, n (%)	47 (94)	48 (94.1)	1.000
Prior AMI, n (%)	21 (42)	24 (47.1)	0.609
Stroke, n (%)	2 (4)	1 (2)	0.617
Smoking, n (%)	10 (20)	14 (27.5)	0.379
Echocardiographic parameters			
LVEF, %	59.5 (56; 64)	60 (55; 67)	0.276
LVESD/BSA, cm/m^2^	16.5 (15.7; 18.4)	16 (14.6; 17.6)	0.138
LVEDD/BSA, cm/m^2^	24.8 (23.2; 26.7)	23.6 (21.8; 26.1)	0.135
LVESV/BSA, mL/m^2^	22.8 (19.2; 28.6)	20.4 (16.3; 26.1)	0.058
LVEDV/BSA, mL/m^2^	56.3 (48.8; 69.2)	53.3 (44.2; 62.4)	0.079
AV peak gradient, mm Hg	9 (7; 47.3)	9.5 (5.2; 44.8)	0.711
AV mean gradient, mm H g	5 (3.4; 31.5)	10.5 (3.9; 55)	0.171
MR, degree	1.5 (1; 2)	1.5 (1; 1.5)	0.483
AR, degree	1 (0; 1,5)	1.5 (1; 1.5)	0.122
LV PWth, mm	9 (9; 14.3)	13 (11.5; 15)	0.211
IVS, mm	13 (12; 15)	13 (12; 14.3)	0.935
LA size, cm	4.2 (3.8; 4.6)	4.1 (3.7; 4.5)	0.596
Laboratory data			
WBC, 10^9^/L	7.9 (6.8; 8.7)	7.5 (6.6; 9.1)	0.973
Neutrophils, 10^9^/L	4.6 (3.8; 5.2)	4.7 (3.5; 5.3)	0.863
Neutrophils, %	60.7 (54; 65.1)	58.8 (52.5; 63)	0.532
Platelets,10^9^/L	247.5 (191.7; 295)	253 (205; 292)	0.519
Creatinine, mkmol/L	85 (73; 94.3)	80 (74; 95)	0.732
eGFR mL/min per 1.73 m^2^ (MDRD)	93.5 (79.8; 107.3)	90 (77; 109)	0.954
Glucose, mmol/L	5.6 (5; 6.5)	5.5 (5; 6.1)	0.530
AST, IU/L	20.5 (17; 30.8)	21 (17.8; 28.5)	0.921
ALT, IU/L	23 (18; 33)	27 (17.8; 34)	0.444
Potassium, mmol/L	4.4 (4.1; 5)	4.6 (4.2; 4.8)	0.488
Drug therapy			
Beta-blockers, %	41 (82)	37 (72.5)	0.257
ACE inhibitors, %	31 (62)	29 (56.9)	0.599
Calcium antagonists, %	17 (34)	16 (31.4)	0.778
Thiazide diuretics,%	8 (16)	3 (5.9)	0.122
Loop diuretics, %	7 (14)	2 (3.9)	0.092
Potassium-sparing diuretics, %	8 (16)	8 (15.7)	1.000
NSAIDs, n (%)	0 (0)	0 (0)	
Acetylsalicylic acid, n (%)	11 (22)	8 (15.7)	0.455
Other antiaggregant, n (%)	3 (6)	3 (5.9)	1.000
Nitrates, %	14 (28)	12 (23.5)	0.607
Statins, %	35 (70)	35 (68.6)	0.881
LMWHs/UFH, n (%)	14 (28)	15 (29.4)	0.875

BSA—body surface area, BMI—body mass index, COPD-chronic obstructive pulmonary disease, AMI—acute myocardial infarction, LV EF—left ventricular ejection fraction, LVEDD—left ventricular end diastolic diameter, LVEDV—left ventricular end diastolic volume, LVESD—left ventricular end systolic diameter, LVESV—left ventricular end systolic volume, AV—aortic valve, MR—mitral regurgitation, AR—aortic regurgitation, LV PWth—left ventricular posterior wall thickness, LA—eft atrium, WBC-white blood cells, eGFR—glomerular filtration rate, MDRD—Modification of diet in renal disease, AST—aspartate aminotransferase, ALT—alanine aminotransferase, ACE—angiotensin-converting enzyme, NSAIDs—nonsteroidal anti-inflammatory drugs, LMWHs—low-molecular-weight heparins.

**Table 2 jcm-11-01387-t002:** Operational and postoperative data.

Parameters	Colchicine (n = 50)	Placebo (n = 51)	*p*
Extracorporeal circulation, n (%)	34 (68)	35 (68.6)	0.946
CPB time, min	120 (89; 138)	120 (90; 142)	0.736
Cardioplegia, n (%)	16 (32)	14 (27.5)	0.617
ACC time, min	63.5 (59.3; 71.5)	67.5 (53.8; 80)	0.755
CABG, n (%)	40 (80)	45 (88.2)	0.288
AV repair, n (%)	16 (32)	13 (25.5)	0.470
Cardiotonic support in ICU, n (%)	19 (38)	15 (29.4)	0.361
Lung ventilation time, h	8 (5.8; 13.3)	8.2 (5.2; 14.4)	0.911

CPB—cardiopulmonary bypass, ACC—aortic cross-clamp, CABG—coronary artery bypass grafting, AV—aortic valve, ICU—intensive care unit.

**Table 3 jcm-11-01387-t003:** Clinical outcomes and complications.

Parameters	Colchicine (n = 50)	Placebo (n = 51)	OR	95% CI	*p*
POAF, n (%)	9 (18)	15 (29.4)	0.527	0.206–1.349	0.178
Effective management of POAF (AAT, cardioversion), n (%)	9 (18)	15 (29.4)	0.527	0.206–1.349	0.178
Ineffective management of POAF, n (%)	0 (0)	0 (0)			
Hospital mortality, n (%)	0 (0)	0 (0)			
Complications in the department:					
Respiratory failure, n (%)	0 (0)	0 (0)			
TIA, n (%)	0 (0)	0 (0)			
Bleeding, n (%)	0 (0)	0 (0)			
General infectious complications, n (%)	0 (0)	0 (0)			
Infectious complications of postoperative wound, n (%)	0 (0)	0 (0)			
Acute renal failure (3 postoperative day)	1 (2)	3 (5.9)	0.327	0033–3.250	0.618
Acute renal failure (5 postoperative day)	1 (2)	0 (0)			1.000
Arrhythmias, except AF, n (%)	4 (8)	3 (5.9)	1.524	0.322–7.202	0.715
PVC, n (%)	0 (0)	0 (0)			
SVESs, n (%)	0 (0)	1 (2)			1.000
AV-block, n (%)	2 (4)	0 (0)			0.243
Others, n (%)	2 (4)	2 (3.9)	1.021	0.138–7.543	1.000
Pacemaker implantation, n (%)	1 (2)	0 (0)			0.495

POAF—postoperative atrial fibrillation, AAT—antiarrhythmic therapy, TIA—transient ischemic attack, PVC—premature ventricular complex, SVESs—supraventricular extrasystoles, AV-block—atrioventricular block, OR—odds ratio, CI—confidence interval.

**Table 4 jcm-11-01387-t004:** Postoperative echocardiographic parameters.

Parameters	Colchicine (n = 50)	Placebo (n = 51)	*p*
postoperative day 3			
LVEF, %	54 (52; 56)	55 (53; 56)	0.221
LVESV/BSA mL/m^2^	21.5 (17.6; 29.4)	18.7 (16.2; 23.1)	0.051
LVEDV/BSA mL/m^2^	47.7 (43.1; 57.6)	43.8 (35.8; 52.6)	0.032
Pericardial effusion, n (%)	3 (6)	8 (15.7)	0.204
Pericardial effusion, mm	5 (4; 5)	5 (5; 10.8)	0.220
Pericardial effusion, n (%)	22 (44)	16 (31.4)	0.139
Pericardial effusion, mm	19 (11; 21.5)	19 (10; 27)	0.811
postoperative day 5			
LVEF, %	55 (51; 56)	55 (54.2; 57)	0.098
LVESV/BSA mL/m^2^	21.5 (17.9; 26.6)	18.5 (16.3; 23.9)	0.075
LVEDV/BSA mL/m^2^	49.3 (42.3; 57.6)	42.8 (37.2; 53.3)	0.027
Pericardial effusion, n (%)	8 (16)	15 (29,4)	0.094
Pericardial effusion, mm	5.5 (4.25; 6)	5 (4; 5.5)	0.427
Pericardial effusion, n (%)	19 (38)	21 (41.2)	0.809
Pericardial effusion, mm	15 (11.5; 22)	20 (12.3; 33.5)	0.225

LV EF—left ventricular ejection fraction, LVESV—left ventricular end systolic volume, BSA—body surface area, LVEDV—left ventricular end diastolic volume.

**Table 5 jcm-11-01387-t005:** Postoperative laboratory data.

Parameters	Colchicine (n = 50)	Placebo (n = 51)	*p*
postoperative day 3			
WBC, 10^9^/L	12 (9.5; 14.4)	11.8 (10.1; 14)	0.869
Neutrophils, 10^9^/L	10.4 (6.5; 12)	8.8 (7.2; 13)	0.965
Neutrophils, %	80 (68; 85.3)	77.5 (70.6; 84.8)	0.811
Platelets, 10^9^/L	201.7 (143; 251.5)	206.5 (171.9; 277.9)	0.324
Creatinine, mkmol/L	75.7 (67; 85.5)	75.9 (68,3; 90.5)	0.415
eGFR mL/min per 1.73 m^2^ (MDRD)	106 (87.8; 123.3)	98.5 (85.8; 120.8)	0.318
Glucose, mmol/L	7.4 (5.4; 8.6)	7 (5.7; 8.2)	0.771
AST, IU/L	31 (24; 43)	28 (20.5; 39)	0.407
ALT, IU/L	20 (15; 30)	23 (15; 30.5)	0.812
Potassium, mmol/L	4 (3.9; 4.5)	4.3 (3.9; 4.7)	0.070
postoperative day 5			
WBC, 10^9^/L	9.1 (8.1; 10.5)	9.7 (8.1; 11)	0.417
Neutrophils, 10^9^/L	5.8 (4.8; 7)	6.3 (5; 7.2)	0.559
Neutrophils, %	61.5 (58.2; 67.2)	63.4 (56.5; 66)	1.000
Platelets, 10^9^/L	255 (194.8; 315)	262 (213; 307)	0.573
Creatinine, mkmol/L	76 (71; 86.4)	73.4 (67.6; 92.1)	0.607
eGFR mL/min per 1.73 m^2^ (MDRD)	100 (83; 111)	98 (79.8; 129.8)	0.889
Glucose, mmol/L	6.6 (5.3; 7.5)	6.1 (5.6; 8.6)	0.820
AST, IU/L	26 (21; 31)	30 (20; 46)	0.215
ALT, IU/L	26 (17; 37)	28 (21; 45)	0.218
Potassium, mmol/L	4.1 (3.8; 4.4)	4.3 (4; 4.8)	0.248

WBC—white blood cells, eGFR—glomerular filtration rate, MDRD—modification of diet in renal disease, AST—aspartate aminotransferase, ALT—alanine aminotransferase.

**Table 6 jcm-11-01387-t006:** Drug therapy in the postoperative period.

Parameters	Colchicine (n = 50)	Placebo (n = 51)	*p*
postoperative day 1			
Beta-blockers, n (%)	46 (92)	44 (86.3)	0.525
Statins, n (%)	35 (70)	35 (68.6)	0.881
ACE inhibitors, n (%)	30 (60)	31 (60.8)	0.936
Calcium antagonists, n (%)	15 (30)	12 (23.5)	0.463
Thiazide diuretics, n (%)	5 (10)	4 (7.8)	0.741
Loop diuretics, n (%)	3 (6)	2 (3.9)	0.678
Potassium-sparing diuretics, n (%)	22 (44)	11 (21.6)	0.016
NSAIDs, n (%)	20 (40)	15 (29.4)	0.264
Acetylsalicylic acid, n (%)	41 (82)	40 (78.4)	0.804
Other antiaggregant, n (%)	35 (70)	41 (80.4)	0.226
Nitrates, n (%)	3 (6)	8 (15.7)	0.200
Antiarrhythmic drugs, n (%)	3 (6)	4 (7.8)	1.000
Cardiotonic support, n (%)	28 (56)	29 (56.9)	0.930
Adrenaline, n (%)	3 (6)	2 (3.9)	0.678
Norepinephrine, n (%)	14 (28)	11(21.6)	0.454
Dopamine, n (%)	19 (38)	25 (49)	0.264
Dobutamine, n (%)	0 (0)	0 (0)	
LMWHs/UFH, n (%)	50 (100)	51 (100)	
Warfarin, n (%)	16 (32)	12 (23.5)	0.342
Antibiotics, n (%)	49 (98)	47 (92.2)	0.362
Steroids, n (%)	38 (76)	41 (80.4)	0.593
postoperative day 3			
Beta-blockers, n (%)	44 (88)	45 (88.2)	1.000
Statins, n (%)	36 (72)	36 (70.6)	0.875
ACE inhibitors, n (%)	29 (58)	32 (62.7)	0.626
Calcium antagonists, n (%)	14 (28)	10 (19.6)	0.322
Thiazide diuretics, n (%)	4 (8)	4 (7.8)	1.000
Loop diuretics, n (%)	5 (10)	5 (9.8)	1.000
Potassium-sparing diuretics, n (%)	24 (48)	17 (33.3)	0.133
NSAIDs, n (%)	20 (40)	16 (31.4)	0.365
Acetylsalicylic acid, n (%)	39 (78)	41 (80.4)	0.767
Other antiaggregant, n (%)	35 (70)	40 (78.4)	0.333
Nitrates, n (%)	1 (2)	2 (3.9)	1.000
Antiarrhythmic drugs, n (%)	4 (8)	6 (11.8)	0.741
Cardiotonic support, n (%)	14 (28)	14 (27.5)	0.951
Adrenaline, n (%)	1 (2)	2 (3.9)	1.000
Norepinephrine, n (%)	9 (18)	5 (9.8)	0.263
Dopamine, n (%)	8 (16)	11 (216)	0.612
Dobutamine, n (%)	0 (0)	0 (0)	
LMWHs/UFH, n (%)	48 (96)	45 (88.2)	0.269
Warfarin, n (%)	16 (32)	14 (27.5)	0.617
Antibiotics, n (%)	36 (72)	39 (76.5)	0.607
Steroids, n (%)	17 (34)	18 (35.3)	0.891
postoperative day 5			
Beta-blockers, n (%)	44 (88)	43 (84.3)	0.775
Statins, n (%)	36 (72)	36 (70.6)	0.875
ACE inhibitors, n (%)	30 (60)	31 (60.8)	0.936
Calcium antagonists, n (%)	13 (26)	10 (19.6)	0.444
Thiazide diuretics, n (%)	4 (8)	3 (5.9)	0.715
Loop diuretics, n (%)	3 (6)	6 (11.8)	0.487
Potassium-sparing diuretics, n (%)	24 (48)	17 (33.3)	0.133
NSAIDs, n (%)	15 (30)	7 (13.7)	0.048
Acetylsalicylic acid, n (%)	42 (84)	42 (82.4)	1.000
Other antiaggregant, n (%)	36 (72)	41 (80.4)	0.322
Nitrates, n (%)	0 (0)	2 (3.9)	0.495
Antiarrhythmic drugs, n (%)	6 (12)	8 (15,7)	0.775
Cardiotonic support, n (%)	2 (4)	1 (2)	0.617
Adrenaline, n (%)	1 (2)	0 (0)	0.495
Norepinephrine, n (%)	0 (0)	0 (0)	
Dopamine, n (%)	1 (2)	1 (2)	1.000
Dobutamine, n (%)	0 (0)	0 (0)	
LMWHs/UFH, n (%)	40 (80)	38 (74.5)	0.511
Warfarin, n (%)	16 (32)	15 (29.4)	0.778
Antibiotics, n (%)	23 (46)	32 (62.7)	0.091
Steroids, n (%)	4 (8)	6 (11.8)	0.741

ACE—angiotensin-converting enzyme, NSAIDs—nonsteroidal anti-inflammatory drugs, LMWHs—low-molecular-weight heparins.

**Table 7 jcm-11-01387-t007:** Adverse clinical events.

Parameters	Colchicine (n = 50)	Placebo (n = 51)	OR	95% Cl	*p*
Nausea, n (%)	6 (12)	5 (9.8)	1.255	0.357–4.408	0.755
Vomiting, n (%)	1 (2)	2 (3.9)	0.500	0.044–5.696	1.000
Lack of appetite, n (%)	11 (22)	14 (27.5)	0.745	0.3–1.85	0.605
Diarrhea, n (%)	16 (32)	6 (11.8)	3.529	1.249–9.972	0.010
Abdominal pain, n (%)	6 (12)	2 (3.9)	3.341	0.641–17.419	0.151
Convulsions, n (%)	1 (2)	4 (7.8)	0.24	0.026–2.225	0.363
Tingling in the extremities, n (%)	5 (10)	7 (13.7)	0.698	0.206–2.367	0.761
Skin rashes, n (%)	0 (0)	0 (0)			

## Data Availability

The datasets analyzed during the current study are publicly available. The data is available in the general repository “Open Science Framework” at the link https://doi.org/10.17605/OSF.IO/HPCZW (accessed on 3 February 2022).

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
