# Peer review of "Colchicine for Prevention of Atrial Fibrillation after Cardiac Surgery in the Early Postoperative Period"

_jcm, 2022, doi:10.3390/jcm11051387_

Round 1
Reviewer 1 Report
In this RCT, Dr. Shvartz and colleagues investigated the effect of colchicine for the prevention of POAF. Overall, this is a nicely conducted study with a clear clinical insight. Nonetheless, some issues were found and need to be clarified by the authors:
- The writing needs to be improved, especially section 2.7 explaining the secondary outcome of the study. Please make sure that the grammatical errors are corrected.
- "The rhythm control was carried out immediately after the operation and until the end of day 7 in the postoperative period." What do the authors mean by "rhythm control"? pharmacological rhythm control with antiarrhythmics? Please make it clear.
- Line 51: "certain role of inflammatory processes in the pathogenesis of POAF" Please include this study as well (PMID: 32762493) since they showed the involvement of IL-1b and NLRP3 inflammasome in POAF.
- Regarding section 2.3 about the exclusion criteria of the study, I am curious why the authors did not exclude the use of NSAIDs before and after surgery. They could influence the observation and affect the results. If this was not done before, please include it as the limitation of the study (commonly located at the end of the discussion section).
- "2-leaf aortic valve" can be phrased as "bicuspid aortic valve"?
- In Section 2.6, "Number of participants with POAF" the intention of this incomplete sentence was not clear. Was it the primary endpoint? Then please make a complete sentence so it is clear to the readers.
- Also, the authors need to describe the diagnostic criteria of POAF they used in this study. What did they see in the ECG strip to confirm the diagnosis of AF post surgery? Please explain in Section 2.6
- Section 2.7 needs to be improved. All the secondary endpoints need to be specified in a clear full sentence (not only italicized phrases).
- "Main lethal and nonlethal nosocomial events are (death, stroke, myocardial infarction, congestive heart failure)."
- Line 122: "Biomarker dynamics of inflammation in blood plasma (neutrophils)." what does this mean? Please clarify.
- Line 125: "Biomarker dynamics of liver damage: aspartate aminotransferase (AST), alanine aminotransferase (ALT)." Make a complete sentence please. Also, what does it mean by "biomarker dynamics"? How dynamic was it? Please elaborate or revise.
- Line 159: "this was the case with just one patient" please describe "this". The authors need to repeat the statement about extended stay in ICU in this sentence.
- "Figure 1. CONSORT diagramma."
- In all tables, please make sure to change the decimal sign from "," to ".". For example, p-value of 0.420.
- The discussion should be extended (the current discussion is not adequate). If it was not due to small sample size, what else could affect the insignificance? Please compare with previous RCTs, real world data and also meta-analyses about colchicine and POAF. Maybe the doses of colchicine or the administration approach were different in this study? Or differences in baseline characteristics of the cohort? In general, please analyze all the possibilities that can justify the insignificance or the possible cause of this shortcoming.
- The authors need to also specifically discuss COPPS-2 and check why they didn't see any difference. Is there any similarity with the current observation?
- The limitations of the study need to be grouped at the end of the discussion section (if not in a separate subsection). Please reorganize it that way to make it easier to identify the limitations.
- Also, do the authors think that assessing neutrophil is enough? Perhaps more inflammatory markers need to be assessed (IL-1, IL-6, IL-10, TNFa, NLRP3 etc)? If so, please mention this in the limitation section as well.
- In the conclusion, "The frequency of POAF in the treatment group was 18% versus 29.4% in the control group, but the detected differences were not statistically significant (OR 0.527; 95% Cl 0.206-1.349; p = 0.178)." although this statement is true, please also add the likelihood of the inaccuracy due to small (inadequate) sample size.
- Have the authors tried to calculate the sample size using the approach shown here (PMID: 22263004)? Please check it and see if the result is consistent with the number generated by the Lehr formula.
Author Response
Response to Reviewer 1
In this RCT, Dr. Shvartz and colleagues investigated the effect of colchicine for the prevention of POAF. Overall, this is a nicely conducted study with a clear clinical insight. Nonetheless, some issues were found and need to be clarified by the authors
We are grateful to our reviewer for his or her very important comments to our work. We tried to correct all shortcomings according to these comments. All changes in the text are highlighted in green.
The writing needs to be improved, especially section 2.7 explaining the secondary outcome of the study. Please make sure that the grammatical errors are corrected.
We consider that your remark is correct and we have made some explanations in the article about obtained indicators.
The primary end point was the frequency of POAF in groups within 7 days after surgery. The An episode with the absence of visible regular P-waves, the appearance of f-waves and irregular R-R intervals on the ECG for more than 5 minutes was the diagnostic confirmation of POAF.
The secondary end points were the number of patients with lethal and nonlethal events, pericardial effusion, pleural effusion, acute kidney injury (AKI), plasma inflammation and liver damage within 7 days after surgery. The main lethal and non-lethal nosocomial events were death, stroke, myocardial infarction and congestive heart failure. The pericardial and pleura effusion was assessed by the echocardiography. The definition of AKI was based on KDIGO criteria (occurrence of AKI was defined as either an increase in serum creatinine ≥0.3 mg/dL within 48 hours or an increase to ≥1.5 times baseline within 7 days). The marker of plasma inflammation was the number of leukocytes and neutrophils. The liver damage was assessed by the level of aspartate aminotransferase (AST) and alanine aminotransferase (ALT).
We have made changes in the article according to your comment.
"The rhythm control was carried out immediately after the operation and until the end of day 7 in the postoperative period." What do the authors mean by "rhythm control"? pharmacological rhythm control with antiarrhythmics? Please make it clear.
By «rhythm control», we meant daily ECG monitoring after surgery to identify the primary endpoint - POAF. We have already changed this part in the article where we clarified that the primary endpoint was the recorded episode of AF which lasted for more than 5 minutes within the first 7 days after surgery.
We have made changes in the article according to your comment.
Line 51: "certain role of inflammatory processes in the pathogenesis of POAF" Please include this study as well (PMID: 32762493) since they showed the involvement of IL-1b and NLRP3 inflammasome in POAF.
We have added the link to this study in the article.
Heijman, J.; Muna, A.P.; Veleva, T.; Molina, C.E.; Sutanto, H.; Tekook, M.; Wang, Q.; Abu-Taha, I.H.; Gorka, M.; Künzel, S.; El-Armouche, A.; Reichenspurner, H.; Kamler, M.; Nikolaev, V.; Ravens, U.; Li, N.; Nattel, S.; Wehrens, X.H.T.; Dobrev, D. Atrial Myocyte NLRP3/CaMKII Nexus Forms a Substrate for Postoperative Atrial Fibrillation. Circ Res. 2020, 127(8), 1036-1055.
Regarding section 2.3 about the exclusion criteria of the study, I am curious why the authors did not exclude the use of NSAIDs before and after surgery. They could influence the observation and affect the results. If this was not done before, please include it as the limitation of the study (commonly located at the end of the discussion section).
We agree, that taking the anti-inflammatory drugs (NSAIDs) and steroids has an effect on inflammation and can affect the frequency of POFP. In the exclusion criteria of this study, we identified that the taking steroids in preoperative period was the exclusion criteria and also mentioned that we also exclude various inflammatory conditions. Therefore, there was no use of NSAIDs in our study before surgery (Table 1). As for the postoperative period, about 50% of patients receive NSAIDs for the purpose of analgesia to avoid the pain syndrome after open surgery. In this regard, adding NSAIDs to the exclusion criteria in the postoperative period would significantly reduce the sample for our study. And taking into consideration the randomized nature of the study, and the effects of NSAIDs evenly on both groups of patients in the postoperative period, we think that this is not an important limitation of the study. However, on the recommendation of the reviewer, we have added this information to the limitations of the study section.
"2-leaf aortic valve" can be phrased as "bicuspid aortic valve"?
We have changed "2-leaf aortic valve" to "bicuspid aortic valve" in the article.
In Section 2.6, "Number of participants with POAF" the intention of this incomplete sentence was not clear. Was it the primary endpoint? Then please make a complete sentence so it is clear to the readers.
Thank you for your comment.
Initially, in this article, we described the primary and secondary endpoints in that formulations as they were described in the protocol in Clinical Trials Registration: NCT04224545, which the reviewers of this resource recommended to us.
We agree, that it may not be clear to readers. And we have corrected these formulations in sections 2.6 and 2.7.
Also, the authors need to describe the diagnostic criteria of POAF they used in this study. What did they see in the ECG strip to confirm the diagnosis of AF post surgery? Please explain in Section 2.6
We have made changes in the article according to your comment.
An episode with the absence of visible regular P-waves, the appearance of f-waves and irregular R-R intervals on the ECG for more than 5 minutes was the diagnostic confirmation of POAF.
Section 2.7 needs to be improved. All the secondary endpoints need to be specified in a clear full sentence (not only italicized phrases). "Main lethal and nonlethal nosocomial events are (death, stroke, myocardial infarction, congestive heart failure)."
We have indicated that secondary end points were the frequency of lethal and nonlethal events, pericardial effusion, pleural effusion, acute kidney injury (AKI), plasma inflammation markers, liver damage within 7 days after surgery. The main lethal and non-lethal nosocomial events were death, stroke, myocardial infarction, congestive heart failure. The pericardial and pleura effusion was assessed by echocardiography. The definition of AKI was based on KDIGO criteria (occurrence of AKI was defined as either an increase in serum creatinine ≥0.3 mg/dL within 48 hours or an increase to ≥1.5 times baseline within 7 days). The plasma inflammation markers were the number of leukocytes and neutrophils. The liver damage was assessed by the level of aspartate aminotransferase (AST) and alanine aminotransferase (ALT).
We have made changes in the article.
Line 122: "Biomarker dynamics of inflammation in blood plasma (neutrophils)." what does this mean? Please clarify.
By "biomarker dynamics" we meant statistical differences between the study group and the control group in the postoperative period in terms of the level of WBC, neutrophils on the 3rd and 5th postoperative days.
We made edits to this section: "The markers of blood plasma inflammation were the number of leukocytes and neutrophils."
Section 2.9 says that on the 3rd and 5th days after the surgery, all participants underwent laboratory tests (general blood test, biochemical blood test).
We have made changes in the article according to your comment.
Line 125: "Biomarker dynamics of liver damage: aspartate aminotransferase (AST), alanine aminotransferase (ALT)." Make a complete sentence please. Also, what does it mean by "biomarker dynamics"? How dynamic was it? Please elaborate or revise.
By "biomarker dynamics" we meant statistical differences between the study group and the control group in the postoperative period in terms of the level of aspartate aminotransferase and alanine aminotransferase on the 3rd and 5th postoperative days.
We have also corrected these formulations.
Line 159: "this was the case with just one patient" please describe "this". The authors need to repeat the statement about extended stay in ICU in this sentence.
We agree with your comment. We changed "it was only with one patient" to "one patient dropped out from the study due to extended stay at the intensive care unit ".
"Figure 1. CONSORT diagramma."
We have made changes to the name of Figure 1.
In all tables, please make sure to change the decimal sign from "," to ".". For example, p-value of 0.420.
We changed the sign "," to "." in all tables.
The discussion should be extended (the current discussion is not adequate). If it was not due to small sample size, what else could affect the insignificance? Please compare with previous RCTs, real world data and also meta-analyses about colchicine and POAF. Maybe the doses of colchicine or the administration approach were different in this study? Or differences in baseline characteristics of the cohort? In general, please analyze all the possibilities that can justify the insignificance or the possible cause of this shortcoming. The authors need to also specifically discuss COPPS-2 and check why they didn't see any difference. Is there any similarity with the current observation?
Thank you for your comment.
We have supplemented the «Discussion» section and compared our study with previous studies.
In COPPS-1, patients showed a significant decrease in POPF after open-heart surgery. However, in COPPS-2, colchicine did not have a significant effect on the incidence of POAF. Based on the results of the meta-analysis, a decrease in the frequency of POAF was revealed in the group of patients receiving colchicine compared with the control group.
Contradictory research results may be associated with different inclusion and exclusion criteria, the duration of colchicine intake, the period of patient follow-up and the diagnostic criteria of POAF.
In the COOPS-2 study, colchicine was administered within 48-72 hours before surgery and within 1 month after surgery. In our study, patients take colchicine once a day before surgery and on the 2nd, 5th days after surgery. The primary endpoint in the COOPS-2 study was an episode of POAF lasted at least 30 seconds, recorded within 3 months after surgery. In our study, the observation of patients was limited to 7 postoperative days, and the diagnostic criteria for AF was an episode of AF lasted at least 5 minutes. This really affects the different frequency of primary endpoint detection in our and previous studies. It is logical that the stricter the criteria for the diagnosis of POAF, the frequency of detection of the event will be higher. However, at the moment there are no clear universal criteria for determining clinical POAF (30 seconds or 5 minutes). There are several studies where POAF has been defined as AF lasted 10 minutes and more. Also, there is no reliable data on prognostic differences in events with different interpretations of the definition of POFP.
In the COOPS-1 study, colchicine intake was carried out on the 3rd day after surgery and lasted for 1 month after surgery. The patients with episodes of AF registered on the 1st and 2nd postoperative days were excluded from the analysis, while the episode of AF with duration at least of 5 minutes as well as in our study was considered to be the diagnostic criteria.
Also, in the COOPS-1 and COOPS-2 studies, patients with a history of AF, taking antiarrhythmic drugs in the preoperative period, mitral valve defects, which have been proven to increase the risk of AF were not excluded initially from the study. In addition, these studies included patients who underwent CABG, operations on various heart valves and aorta. Our study included patients only after CABG surgery and/or replacement of aortic valve (without significant mitral pathology), and the atrial fibrillation or atrial flutter in history was the exclusion criteria for our study. Thus, we initially excluded patients with a high risk of developing of AF in order to minimize the influence of existing predictors of occurrence of AF in the preoperative period. We consider that our cohort was more homogeneous initially, although it was smaller than in previous studies.
The limitations of the study need to be grouped at the end of the discussion section (if not in a separate subsection). Please reorganize it that way to make it easier to identify the limitations.
We added the section "Limitations of the Study" to the article and indicated possible limitations of our research.
“Limitations of the Study”
In this study, there was a small sample of patients, although it was more homogeneous, than in previous studies.
Also, in the study, some patients took NSAIDs in the postoperative period, that could affect the results.
For the assessment of inflammation in the postoperative period the levels of leukocytes and neutrophils in the blood were studied as the markers of inflamation, although the markers of the interleukin family (IL-1, IL-6, IL-10), TNFa, NLRP3, etc. are more specific.
Also, do the authors think that assessing neutrophil is enough? Perhaps more inflammatory markers need to be assessed (IL-1, IL-6, IL-10, TNFa, NLRP3 etc)? If so, please mention this in the limitation section as well.
Indeed, we believe that markers of the interleukin family (IL-1, IL-6, IL-10) are more specific than neutrophils or leukocytes. However, these analyses are not included in our routine clinical practice. Therefore, we used more cost-saving and routine option. We focused in our study on a clinical phenomenon - the development of AF, which was our primary endpoint, but the inflammation was the secondary one. We have indicated this in the limitations of the study.
In the conclusion, "The frequency of POAF in the treatment group was 18% versus 29.4% in the control group, but the detected differences were not statistically significant (OR 0.527; 95% Cl 0.206-1.349; p = 0.178)." although this statement is true, please also add the likelihood of the inaccuracy due to small (inadequate) sample size.
We indicated that the differences didn’t have any statistical significance (OR 0.527; 95% Cl 0.206-1.349; p = 0.178), that could be due to the insufficient volume of the sample of the study.
Have the authors tried to calculate the sample size using the approach shown here (PMID: 22263004)? Please check it and see if the result is consistent with the number generated by the Lehr formula.
Thank you very much for your comments and recommendation. Using the formula for statistical "superiority design" proposed in the article PMID: 22263004, we calculated that there should be 303 patients in each group to obtain statistically significant differences taking into account the frequency of AF occurrence (in the control group 29.4% and 18% in the study group), with a given study power of 80% and a= 0.05. Using the Lehr formula, the estimated number of patients should be at least 223.
There are many different formulas for calculating the desired sample, each of which certainly has its own characteristics, as well as advantages and disadvantages. However, according to the recommendation of another reviewer, we removed the calculation of the required sample based on our own results.

Reviewer 2 Report
There are already several studies on this subject and with the number of patients selected, the study by itself does not solve the question since the result is not significant with an insufficient number of patients. The only added value of this analysis is that it provides about 100 further randomized patients for a potential future meta-analysis.
Methodologically speaking, the authors should:
- either include in Methods a calculation for the needed sample size in order to have enough statistical power based on the expected results according to previous studies on the subject, and explain in Results why such a sample size was not reached,
- or state they use a convenience sample by taking their available patients, instead of mentioning in the discussion the number of patients that would have been necessary to reach statistical significance. This is not to be done a posteriori based on their own results.
In Discussion, the authors should comment on the existing difficulties to perform larger studies and on the need for meta-analyses. They should mention the main contribution of their study is to provide patients data for them.
In Conclusions, when stating the differences were not statistically significant, the authors should add this does not goes to suggest differences do not exist, because with their sample size a non-significant result was in fact to be expected even in the presence of a true difference.
Author Response
Response to Reviewer 2
There are already several studies on this subject and with the number of patients selected, the study by itself does not solve the question since the result is not significant with an insufficient number of patients. The only added value of this analysis is that it provides about 100 further randomized patients for a potential future meta-analysis.
We are grateful to our reviewer for his or her very important comments to our work. We tried to correct all shortcomings according to these comments. All changes in the text are highlighted in green.
Methodologically speaking, the authors should:
- either include in Methods a calculation for the needed sample size in order to have enough statistical power based on the expected results according to previous studies on the subject, and explain in Results why such a sample size was not reached,
- or state they use a convenience sample by taking their available patients, instead of mentioning in the discussion the number of patients that would have been necessary to reach statistical significance. This is not to be done a posteriori based on their own results.
We partly agree with the reviewer, so we have changed the discussion and conclusion. And also, according to his second recommendation, we removed the calculation of the required sample based on our own results.
However, we would like to comment the following:
The existing number of studies with colchicine are extremely small and they are heterogeneous both in terms of the definitions of POAF as well as in the duration of postoperative follow-up period. They are also heterogeneous concerning the inclusion and exclusion criteria, which in our opinion is extremely important.
Also, in the COOPS-1 and COOPS-2 studies, patients with a history of AF, taking antiarrhythmic drugs in the preoperative period, mitral valve defects, which have been proven to increase the risk of AF were not excluded initially from the study. In addition, these studies included patients who underwent CABG, operations on various heart valves and aorta. Our study included patients only after CABG surgery and/or replacement of aortic valve (without significant mitral pathology), and the atrial fibrillation or atrial flutter in history was the exclusion criteria for our study. Thus, we initially excluded patients with a high risk of developing of AF in order to minimize the influence of existing predictors of occurrence of AF in the preoperative period. We consider that our cohort was more homogeneous initially, although it was smaller than in previous studies.
As a result, in our study, with a more homogeneous cohort, the preliminary calculation of the sample size turned out to be difficult, because we could not based on the COPPS-1 and COPPS-2 studies.
In Discussion, the authors should comment on the existing difficulties to perform larger studies and on the need for meta-analyses. They should mention the main contribution of their study is to provide patients data for them.
We agree with your comment. We indicated in the discussion that this study can be considered as a pilot study to provide patient data for subsequent meta-analysis.
In fact, the protocol of our study differs significantly from previous studies by the more homogeneous group of patients and the absence of high risk factors for the development of AF (previous AF in the history, mitral valve interventions, etc.). The results which would be obtained and the data should be used for subsequent meta-analyses and for calculating the required number of patients when planning larger randomized trials in the future.
The datasets analyzed during the current study are publicly available. The data is available in the general repository "Open Science Framework" at the link https://doi.org/10.17605/OSF.IO/HPCZW
In Conclusions, when stating the differences were not statistically significant, the authors should add this does not goes to suggest differences do not exist, because with their sample size a non-significant result was in fact to be expected even in the presence of a true difference.
Thank you very much for your comments.
We have made changes in the article according to your comment.
We indicated that the differences didn’t have any statistical significance (OR 0.527; 95% Cl 0.206-1.349; p = 0.178), that could be due to the insufficient volume of the sample of the study.

Round 2
Reviewer 1 Report
Thank you for the responses. I only have 1 remaining correction: “COOPS” should be “COPPS”. please re read everything again carefully in case there is another typo.
Author Response
We are grateful to our reviewer.
We corrected all typos.
Reviewer 2 Report
.
Author Response
We are grateful to our reviewer.